# Determinants of Continuance Intention towards Banks' Chatbot Services in Vietnam: A Necessity for Sustainable Development

**Dung Minh Nguyen** [1] , **Yen-Ting Helena Chiu** [1] **and Huy Duc Le** [2,*]

1   Department of Marketing and Distribution Management, College of Management, National Kaohsiung University of Science and Technology, Kaohsiung 824005, Taiwan; I108123114@nkust.edu.tw (D.M.N.); helena@nkust.edu.tw (Y.-T.H.C.)
2   UCSI Graduate Business School, UCSI University, Wilayah Persekutuan Kuala Lumpur, Cheras 56000, Malaysia
*   Correspondence: 1002161345@student.ucsiuniversity.edu.my

**Abstract:** To improve customer experience and achieve sustainable development, many industries, especially banking, have leveraged artificial intelligence to implement a chatbot into their customer service. By integrating DeLone and McLean's information systems success (D&M ISS) model and the expectation confirmation model (ECM) with the factor of trust, the aim of this study was to investigate the determinants of users' continuance intentions towards chatbot services in the context of banking in Vietnam. A total of 359 questionnaire surveys were collected from a real bank's chatbot users and analyzed using structural equation modeling. The findings revealed that users' continuance intentions towards the banks' chatbot services were influenced by satisfaction, trust, and perceived usefulness, of which trust had the strongest effect. The results also indicate that information quality, system quality, service quality, and confirmation of expectations had significant effects on three drivers of continuance intention in different ways. Our study contributes to the literature by providing a more comprehensive viewpoint to understand the perceptions and reactions of chatbot users in the post-adoption stage. The results of this study also yield several key suggestions for banking service providers on how to increase their customers' intentions to continue using chatbot services, serving as a basis for long-term and sustainable development strategies in the current digital era.

**Keywords:** chatbot; D&M ISS; ECM; trust; continuance intention; banking

## 1. Introduction

In the current digital transformation era, artificial intelligence (AI) is expected to assist humans with a variety of tasks at work and in their daily lives [1,2]. More remarkably, the outbreak of the COVID-19 pandemic has produced a paradigm shift in the ways we communicate and work, which demonstrates the importance of automated chat functions, particularly chatbots, for various companies' activities [3]. A chatbot, in general, could be understood as AI software that is programmed to automatically communicate with humans via text messages or chats [4]. Currently, chatbot systems are widely used by organizations in many fields, such as customer service, marketing, B2C sales, and training [5,6], to provide their online customers with effective 24/7 service [7]. In addition, digitization has also been transforming the landscapes of various industries [8], especially those of the financial and banking sector with the appearance of the emerging Fintech trend, including various applications, such as online banking, internet cards, digital payments, and cryptocurrencies [9,10]. In fact, difficulties of the current pandemic with face-to-face interactions and mobility accidentally expedited these Fintech-based applications, which help customers to experience the services in a convenient way. The usage of a chatbot in banks, typical financial institutions, is equally worth discussing.

Applying a chatbot to customer service has gained in popularity, benefitting both firms and customers. On the customer side, with traditional customer service, customers usually suffer from queuing and waiting for a response to solve their issues due to a lack of service personnel, which may cause a negative service experience [7,11]. By contrast, virtual agents, such as chatbots, are capable of providing immediate responses and relevant information to customers' problems [3,12]. Additionally, Dospinescu et al. [13] argued that waiting times, transaction costs, and competitive services were the most important factors determining customer satisfaction in the relationship with service providers (i.e., banks). Hence, responsive chatbot-aided customer service is thereby considered the key to customer satisfaction [14]. On the firm side, chatbot services are able to handle a large number of customers' requirements, 24/7, with the absence of employee engagement, enabling firms to effectively reduce operating costs [15]. From a long-term perspective, applications of chatbots, together with other technology-enabled solutions, are expected to enhance the sustainable development of businesses [10]. For these benefits, chatbots are implemented in various industries from banking, retail, and healthcare to tourism and hospitality. According to a report of Grand View Research [16], the chatbot market, estimated at USD 430.9 million in 2020, is expected to reach USD 2.486 million in 2028, progressing at a compound annual growth rate of 24.9% over the 2021–2028 period. The adoption of chatbots is also estimated to save the retail, banking, and healthcare sectors USD 11 billion annually by 2023 [17].

For this study, we placed an emphasis on the chatbot services in Vietnamese banks for several reasons. First, banking is considered one of the typical industries that majorly reaps benefits from the adoption of chatbot services, together with the retail and tourism fields [18]. Juniper Research [19] estimated that, thanks to chatbots, the operating cost saved in banking globally will be USD 7.3 billion by 2023, approximately 35 times higher than what it was in 2019. Second, numerous banks have started their digital transformation journey [20], and chatbots are considered to be essential and indispensable contributors to the transformation as well as sustainable strategies for banking development [21]. Banks often use chatbots in marketing activities, sales, and customer relationship management [22] to provide fast, cost-effective and personalized services to customers. A similar trend can be seen in Vietnamese banks, which are progressing towards the adoption of AI-enabled technology and chatbot services. The report of Austrade [23] showed that by the end of 2019, nearly 60% of commercial banks in Vietnam already had a digital transformation initiative, and more than half of them have implemented chatbots to date. For instance, Tienphong Bank (TPBank) and National Citizen Bank (NCB) are two prominent pioneers of chatbot adoption [24], followed by VPbank, Vietcombank, Techcombank, NamAbank, and Eximbank, whose chatbot systems have also been applied to enhance customer service.

However, although chatbots have been extensively used by many businesses in recent years, customers' satisfaction with chatbots is still rather low. For instance, a recent survey showed that 74% of consumers expect to encounter a chatbot on a website, but only 13% of the surveyed respondents prefer using chatbots over human interactions [25]. This may be due to several issues that arise from chatbot usage, such as uncertainty about the chatbot's performance [26], uncomfortable feelings [27], or privacy concerns [28].

While the degree of users' satisfaction and continuance intentions towards chatbots remains relatively low, very few extant studies have been conducted to investigate why consumers are reluctant to continue using them. For a better understanding of the issue, an empirical examination of the chatbot users' satisfaction and continuance intentions becomes more pertinent and essential, especially for chatbot services in the banking sector. The most recent attempt to investigate customers' satisfaction regarding chatbot services in banking was made by Eren [22]. However, the study did not answer the question of whether or not even the satisfied users will continue using the chatbots in the future. Most importantly, whether the customer is satisfied is the sole variable affecting the likelihood of continuing to use chatbot services, which has also been a fully unanswered question. Hence, drawing on the DeLone and McLean's information systems success (D&M ISS) model,

the expectation confirmation model (ECM), and the trust concept, this study aimed to investigate the key determinants influencing users' continuance intentions towards banks' chatbot services in Vietnam and to explore the process by which these aforementioned effects are created.

The contribution of this study, therefore, is threefold. First, the understanding of the antecedents of chatbot users' continuance intention contributes to the growing literature on the use of chatbots in customer service. Second, by integrating the D&M ISS model and the trust concept into the ECM, this study provides a more comprehensive viewpoint to identify the factors determining the continuance usage intention of chatbot services compared to a single-model analysis, which has not yet been done. Third, the results yielded from this study will help banking service providers and chatbot programmers to better understand the users' reactions after adopting chatbot services and to formulate effective strategies to enhance their continuance usage intention towards chatbots, which contributes to the sustainable development of banks in the long run.

## 2. Literature Review

### 2.1. Chatbot Services

The term "chatbot" is an amalgamation of "chatting" and "robot" [29]. According to Lui and Lam [30], a chatbot is an AI-based computer program that stimulates conversations or interactions with real people through messaging applications and websites. Conversations between humans and chatbots can take place in the form of text-based interactions and spoken interactions without limitations in terms of time and space [31]. Both machine-based interacting forms are dexterously disguised as human agent support, with which users feel more comfortable to start a conversation [32]. The key tasks of the chatbot are to support users in fulfilling information-searching needs, answering queries, and building social relationships [33,34]. Chatbots have been used as firm representatives to provide information value to their customers and satisfy their needs [14,33].

Studies on chatbots have been focused on several aspects. First, conversational systems with speech and chatbot programming methods, referred to as the technical aspect of the chatbots, have been examined [35,36]. Second, several studies have concentrated on user–chatbot communication, such as how chatbot adoption can enhance consumers' purchase intentions [27] and the extent to which customers are willing to adopt the use of and interact with a chatbot [37]. Third, some empirical studies have recently been conducted to explore the issues regarding chatbot adoption in customer service, such as the usability of the chatbot services [38], the effect of chatbot services on customer satisfaction [22,33], and customers' preferences (human vs. chatbot services) in resolving their tasks [39]. These studies have been conducted in various contexts, such as banking services [22], online travel agencies [40], luxury brands [33], and social media [41].

While chatbots play an essential role and have been widely adopted in customer service, not all customers are willing or feel comfortable interacting with them [42]. This may be a reason why user satisfaction has recently received much attention from researchers, as a result of measuring the outcomes of chatbot adoption in customer service. To name a few, Chung et al. [33] found that chatbots with good interactive e-service are able to viably enhance the levels of satisfaction of luxury brand customers. Li et al. [40] examined the relationship between chatbot services and customer satisfaction in the context of online travel agencies and suggested that customer satisfaction could be enhanced when they perceive that the chatbot services are of high quality. However, it is more necessary to answer the question of whether and in which conditions these users who have adopted the use of chatbot services will continue using them in the future. In fact, empirical studies to investigate the key factors influencing customers' intentions to continue using chatbot services have remained limited, especially for chatbot services in the banking sector.

## 2.2. Expectation-Confirmation Model

The main theoretical foundation of the current study is the expectation confirmation model (ECM) proposed by Bhattacherjee [43]. The ECM has its roots in expectation confirmation theory (ECT), which was initially introduced by Oliver [44] and extensively used to evaluate consumer satisfaction for the marketing domain [45,46]. The ECT examines the consumers' behaviors in both pre-consumption and post-consumption stages. The central concept of the ECT is that "satisfaction occurs when expectations are confirmed" [44]. Following that, before using a product/service, consumers tend to form their expectations about it. After using that product/service, consumers will evaluate its performance based on their actual experiences and feelings. By comparing customers' expectations with the performance of the product/service, their expectations are confirmed or disconfirmed, which positively or negatively affects their satisfaction, respectively. The outcomes of such comparisons may influence consumers' satisfaction and repurchasing intentions [43].

However, some scholars have proposed the modified model of the ECT to apply in different research areas due to its insufficient and limited interpretations. For example, Bhattacherjee [43] argued that the ECT [44] ignored the fact that the actual expectations can change over time, and consumers can evaluate their actual expectations during the confirmation stage. Subsequently, Bhattacherjee [43] modified the ETC and proposed the ECM. The ECM inherited two variables from the ECT, including confirmation of expectations and satisfaction. However, the substantial difference between the ECM and the ECT is that while the ECT focuses on pre- and post-consumption factors, the ECM evaluates the related constructs of the post-usage stage [47].

Additionally, the ECM ameliorates the ECT by considering perceived usefulness which represents the post-consumption expectations [48], and the ECM emphasizes the effect of post-consumption expectations rather than that of pre-consumption expectations. Essentially, the ECM posits that an individual tends to continue using an IS after developing expectations about the IS. According to the ECM, confirmation of expectations and perceived usefulness are critical predictors of users' satisfaction, and satisfaction, in turn, will determine their intention to continue using an IS [43]. Bhattacherjee [43] also argued that the ECM is superior to existing models, such as the technology acceptance model [49], the theory of planned behavior [50], the unified theory of acceptance and use of technology [51], for studying the IS continuance behavior since satisfaction and confirmation included in the ECM are more consistent with post-adoption reactions and explanations of the IS continuance.

The ECM model, therefore, is proved to interpret the continuance usage intention successfully, both in information technology and service marketing [42,52,53]. Thus far, the ECM also has played a strong theoretical base to comprehend users' (consumers') continuance and repurchase intentions in various contexts, such as e-magazines [54], mobile advertising [55], mobile payment [56], e-government service [57], and recently, AI-powered service agents [42]. Additionally, a meta-analysis of Ambalov [58] reported that the ECM was a relevant theoretical foundation to examine the satisfaction and continuance intention of IS' users. Thus, we used the ECM as a basis for this empirical study on users' intentions to continue using the banks' chatbot services.

## 2.3. DeLone and McLean's IS Success Model

In this study, DeLone and McLean's IS success model (D&M ISS) [59] comes into play to identify influences of users' satisfaction on intention to continue using the bank's chatbot services. The D&M ISS model, first introduced by DeLone and McLean [60] in 1992, is theoretically sound in explaining the behaviors in the post-adoption stage [56,61]. The original D&M ISS model [60] comprises six constructs defining the successful information systems: information quality, system-based quality, use, users' satisfaction, individual impacts, and organizational impacts. Ten years later, Delone and McLean [59] updated their original model by incorporating a new variable, namely service quality. One year before publishing the updated model, DeLone and his partners [62] explained that the

primary reason for redeveloping their model was the changes in the nature of information systems over time, leading to the change in the notion of "success". Several researchers argued that it is essential to take service quality into account when measuring information systems [63,64]. Hence, with the updated version of the D&M ISS model [59], it is believed that three components of the information systems (i.e., service quality, information quality, system quality) will affect system usage and users' satisfaction, which workably explain the success of the information system platform [56].

After its reinvention, the D&M ISS model [59] has been widely used to evaluate intentions of continuing to use specific information systems. For example, Rahi and Ghani [65] integrated the D&M ISS model into self-determination theory to assess the mutual effects of quality facilitators, users' satisfaction, external motivations, and continuance intention in the context of internet banking. Veeramootoo et al. [57] combined the D&M ISS model, the ETC, habit, and perceived risk to investigate factors that affect the success of e-government services. Hence, it is also well-advised to apply the D&M ISS model as a theoretical framework to understand the users' continuance intentions in the context of banks' chatbot services.

### 2.4. Trust

Ranaweera and Prabhu [66] argued that ''satisfaction'' itself might not be sufficient to maintain a customer's long-term commitment to one specific product/service. Hence, it is necessary to combine satisfaction with other variables, such as trust, to understand customers' repurchase intentions better [67]. Venkatesh et al. [68] also claimed that trust, together with user satisfaction, are the two critical determinants of adoption and continuance intention in e-commerce studies. Thus far, the term "trust" has been studied in various fields (e.g., marketing, psychology, information systems), yet it is still difficult to define and conceptualize the trust concept due to its complicated nature [69].

From a broader perspective, trust can be conceptualized as an individual's belief that other people behave and perform actions within an anticipated range [70]. Since trust could reduce the perceived risk and uncertainty, trust has been considered as one of the crucial elements determining customers' participation in e-commerce [71]. In this study, trust is understood as the degree to which users are confident in the reliability and quality of the chatbot systems [72]. Since chatbots are programmed to perform human-like conversations with users, chatbot users are recommended to consider the potential risks from conversations with chatbots. For example, hackers may create rogue chatbots that impersonate service providers to initiate conversations with users and then convince them to share personal information for malicious purposes. Due to the potential uncertainty and risks, it makes sense to argue that trust is a crucial element influencing users' behavioral intentions towards chatbot services.

Although trust has received much attention in the context of electronic-based services, it is relatively novel in the case of chatbot services [71]. The current study combines trust with the D&M ISS model and the ECM and considers trust as one determinant of users' continuance intentions towards banks' chatbot services.

### 2.5. Integrating ECM, D&M ISS and Trust

Existing research demonstrated that it is possible to integrate the D&M ISS model with other theories or models. For example, Lin et al. [73] combined the D&M ISS model with the UTAUT and the task-technology fit model to indicate how users intend to use mobile payment in Korea. Aldholay et al. [74] combined the D&M ISS model in the context of transformational leadership to evaluate e-learning use. These studies imply that combining the D&M ISS model and other theories would provide a more comprehensive description of the behavioral intentions than the D&M ISS model alone.

On the one hand, the D&M ISS model emphasized the significance of three quality-related dimensions in measuring the whole quality of the information systems and heralding users' satisfaction. However, that model did not consider the user's continuance

intention while satisfaction is a crucial predictor of continuance intention in numerous existing studies (e.g., [57,65,75,76]). On the other hand, while the ECM concentrated on the user's confirmation of expectations of the post-usage stage to predict how likely users are to feel satisfied and continue to use the IS, it did not clarify the factors influencing users' confirmation. Additionally, apart from satisfaction, trust is also a vital factor determining users' continuance intentions [68]. Two of the most recent studies regarding chatbot users' continuance intentions (i.e., [40,42]) found that one of their limitations is omitting the role of trust in users' continuance intentions. With the arguments above, combining all constructs of the D&M ISS model, the ECM, and the trust concept expectedly provides the more comprehensive viewpoint to understand users' continuance intentions in the context of banks' chatbot services.

## 3. Research Model and Hypotheses

In this study, the authors integrated the D&M ISS model, the ECM, and the trust variable to explore the factors influencing users' continuance intentions regarding banks' chatbot services. In detail, hypotheses depicting the relationships among system quality, information quality, service quality, confirmation of expectations, perceived usefulness, trust, satisfaction, and continuance intention were established, in which satisfaction, trust, and perceived usefulness were three determinants of continuance intention. The research framework is shown in Figure 1.

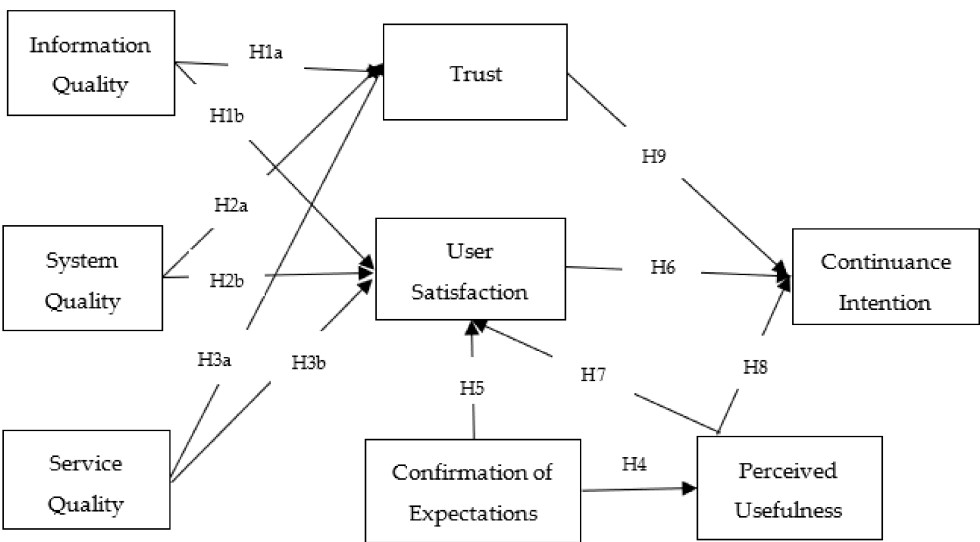

**Figure 1.** Research model.

DeLone and McLean [59] revealed that information quality should be reflected by several characteristics: accuracy, timeliness, integrality, and pertinence. All factors somewhat impact users' satisfaction. Accessing reliable, precise, adequate, and updated information significantly contributes to users' satisfaction [57,77]. Some existing studies also demonstrated information quality as the critical factor stimulating users' trust (e.g., [78–80]). Users spend much time and effort on chatbot services to seek out the information for making decisions. Hence, the information from the chatbot systems should be accurate, straightforward, personalized, and well-presented [77]. Especially since a bank is a financial institution, the information provided by banks must be accurate due to its direct effects on customers' transactions and financial decision-making. If chatbots provide users with irrelevant, outdated, or inaccurate information, users may no longer trust chatbot services and switch to other substitute sources of information. This situation wastes much time and effort of users [75]. Consequently, users may end up having a poor service experience, thereby decreasing their satisfaction. Hence, we proposed that:

**Hypothesis 1a (H1a).** *Information quality positively affects the trust of chatbot users.*

**Hypothesis 1b (H1b).** *Information quality positively affects the satisfaction of chatbot users.*

In our research, system quality reflects the reliability, ease of use, response time, and availability of chatbot systems [59,81]. The system quality of a chatbot could be considered the technical ability of it to provide easy access and instant, reliable information to support users. Poor system quality can reduce user satisfaction since it makes chatbot usage more challenging and will not fulfill chatbot users' needs. Numerous extant studies have demonstrated the positive impact of system quality on user satisfaction (e.g., [59,82–84]). Additionally, prior studies also suggested that the attributes of system quality and the trust concept had some relevance, enabling system quality to predict trust [81,82]. During conversations with chatbots, users are sometimes required to input their private information to serve their needs. Hence, if service providers ensure the reliability and security of chatbot systems, users may have a higher level of trust in their services. Some scholars also argued that if the information systems have a poor interface design that causes difficulties for users, they may not trust service providers' ability in offering high-quality services [81,85]. Accordingly, we propose the following hypotheses:

**Hypothesis 2a (H2a).** *System quality positively affects the trust of chatbot users.*

**Hypothesis 2b (H2b).** *System quality positively affects the satisfaction of chatbot users.*

Thus far, service quality has been considered as one of the traditional determinants of satisfaction. Service quality is defined as the service capability of meeting users' requirements and is reflected by the reliability, assurance, personalization, and service responsiveness [81]. The relationship between service quality and satisfaction was initially explored in marketing and consumer behavior studies [86]. Moreover, the updated D&M ISS model [59] also postulates that good service quality will ensure users are satisfied with the information systems [84,87]. Thus, if chatbots are well-designed to understand users' concerns via prompt and personalized responses, users will perceive high service quality, enhancing their satisfaction. Additionally, service quality was disclosed to affect users' trust [76,81,85]. The instant, reliable, and personalized responses from chatbots can reduce user's time and effort spent on seeking information, positively contributing to their trust. In contrast, the poor service quality, such as interruptions and untimely responses, may cause users to doubt the efficacy of chatbots, consequently reducing user's trust. We, therefore, hypothesize that:

**Hypothesis 3a (H3a).** *Service quality positively affects the trust of chatbot users.*

**Hypothesis 3b (H3b).** *Service quality positively affects the satisfaction of chatbot users.*

Ever since the ECM [43] was successfully proposed to examine users' reactions in the post-acceptance stage and IS continuance, many ECM-based studies in various contexts also found evidence of positive relationships among confirmation of expectations, perceived usefulness, satisfaction, and continuance intention (e.g., [45,57,88,89]). These studies have demonstrated that users' satisfaction was derived from the confirmation of expectations and perceived usefulness of the information systems. In addition, satisfaction and perceived usefulness were two critical determinants of users' continuance intentions. In line with these findings, we argue that the same logic can be applied to the context of chatbot services.

Users may expect to attain some benefits in the chatbot usages, such as time savings, accurate information, and instant support. If the performance of chatbot services meets or exceeds users' prior expectations, users will find that the chatbots are helpful and they will satisfy users' needs. In addition, users' satisfaction after experiencing the chatbot services

will push them to continue using chatbots in the future. Hence, the following hypotheses are proposed:

**Hypothesis 4 (H4).** *Confirmation of expectations positively affects the perceived usefulness of chatbot users.*

**Hypothesis 5 (H5).** *Confirmation of expectations positively affects the satisfaction of chatbot users.*

**Hypothesis 6 (H6).** *Satisfaction positively affects the user's intention to continue using chatbots.*

Davis [49] claimed in his TAM model that perceived usefulness and perceived ease of use are two vital motivational factors influencing user satisfaction and behavioral intentions. Perceived usefulness reflects the users' belief about whether their experiences are enhanced by using a technology [43]. Furthermore, perceived usefulness has been well-substantiated as a determinant of satisfaction and continuance intention in IS services [54,89–91]. Adding to the TAM, Bhattacherjee's ECM [43] suggested that users' satisfaction and continuance intentions towards technological devices are primarily reliant on the extent to which users believe that technology usage can help them perform their tasks effectively. Specifically, suppose users perceive that using chatbot services is helpful for their tasks, such as seeking information or making online transactions. In that case, users' experience with chatbots could be enhanced thanks to prompt responses and practical solutions provided by the chatbots. Consequently, users will feel more satisfied and continue using chatbot services in the future. From the above arguments, it is reasonable to propose the two following hypotheses:

**Hypothesis 7 (H7).** *Perceived usefulness positively affects the satisfaction of chatbot users.*

**Hypothesis 8 (H8).** *Perceived usefulness positively affects the continuance intention of chatbot users.*

Trust plays a crucial role in business relationships in the online environment since gaining trust could reduce risks, worries, and uncertainties [92–94]. By reducing uncertainties, fears, and perceived risks, trust encourages people to participate in e-commerce activities. The extant literature also demonstrated how trust drives both initial behavioral intentions and continuance intentions in various contexts, such as online purchase [92,95], mobile payment [96,97], and Fintech [8].

Based on this evidence, we also expect that trust can contribute to the user's continuance intention towards chatbot usage. Compared to human-based services, using chatbot services is more uncertain and vulnerable, resulting in higher potential risks. For example, users' personal information can be stolen or poorly protected systems can be easily attacked. Hence, when users trust chatbots, they expect to receive reliable services from highly qualified service providers, motivating them to continue using the chatbot. Thus, we propose that:

**Hypothesis 9 (H9).** *Trust positively affects the continuance intention of chatbot users.*

## 4. Methodology

### 4.1. Instrument Design

The questionnaire items of the constructs in this study were adapted from the relevant existing literature. Information quality, with seven items, and system quality, with five items, were modified from Teo et al. [77]. Service quality was modified from Roca et al. [98], with five items. Trust was measured with four items adapted from Gefen et al. [82]. The perceived usefulness scale was obtained from Oghuma et al. [91], with four items. User satisfaction was measured with four items adapted from Teo et al. [77]. Confirmation of expectations scale, with three items, and continuance intention scale, with three items, were adapted from Bhattacherjee [43]. Each item was evaluated on a seven-point Likert scale

ranging from 1 to 7, in which 1 = strongly disagree and 7 = strongly agree. Based on the literature review, the conceptual definitions of all constructs are shown in Table 1.

The questionnaire items were modified to fit the context of the current study. Then, a three-step procedure was conducted to enhance the quality of the measurement items. First, the items adapted from prior studies were translated from English to Vietnamese, and then translated back to English. Second, we invited three doctoral students and two professors who have experience designing questionnaires for the IS-related studies to pretest the first version of the measurement items. The questionnaire was modified based on experts' feedbacks to ensure consistency, comprehensiveness and readability. Third, the pilot test was conducted with 20 respondents who used the banks' chatbot services to ensure the content validity of the measurement items. The final constructs and items are shown in Appendix A, Table A1.

**Table 1.** Conceptual definitions of the constructs.

| Constructs | Definition | Sources |
|---|---|---|
| Information quality | The quality of information and contents provided by chatbot systems | Delone and McLean [59] |
| System quality | The quality of chatbot systems and their technical aspects | Delone and McLean [59] |
| Service quality | Evaluations of the quality of chatbot services in terms of reliability, assurance, responsiveness, personalization | Delone and McLean [59] |
| Trust | The user's level of confidence in the reliability and quality of the bank's chatbot services | Caceres and Paparoidamis [72] |
| Confirmation of expectations | The consistency between the actual outcomes and the users 'expectations towards chatbot services | Bhattacherjee [43]; Chen et al. [45] |
| Perceived usefulness | The extent to which users are confident that using chatbot services can help them finish their tasks efficiently | Davis [49]; Bhattacherjee [43] |
| Satisfaction | The level of user's satisfaction after comparing the actual performance of chatbot service with their expected performance | Oliver [44]; Bhattacherjee [43] |
| Continuance intention | User's intention to continue using chatbot services in the future | Bhattacherjee [43] |

### 4.2. Sample Collection

This study was conducted in Vietnam, which is a developing country. As an emerging market, the adoption of AI-enabled technology and chatbots in service delivery in the Vietnamese banking systems is in progress. As mentioned in the previous section, over one-third of 35 commercial banks in Vietnam currently apply chatbots in their customer service [23]. In addition, Vietnam has a young and golden population structure and a high smartphone ownership ratio; two advantages are conducive to spreading the banks' chatbot services. Therefore, by choosing Vietnamese banks as the research context, this study is expected to serve as the reference for other countries with similar conditions to Vietnam.

The target samples for this study consisted of banks' customers who have experienced the chatbot services of various banks in Vietnam. The data were collected by utilizing a web-based survey platform. In order to enhance the ability to approach the respondents and to increase their awareness of this survey, we shared the questionnaire on social media platforms (Facebook, Zalo, or LinkedIn) and contacted respondents on the fan pages and forums of the banks. To ensure that the survey participants are actual users of the banks' chatbot services, we required them to answer two screening questions: "Have you ever used the chatbot service of the banks?" and "Please write the name of the bank or chatbot which you had experience with". The samples were collected over nearly two months, from November 2020 to early January 2021. We delivered 500 questionnaires in

total, and 447 returned, achieving a response rate of 89.4%. After that, we filtered the responses by considering the answers to the screening questions. As a result, there were only 382 participants who used the banks' chatbot services. Among them, 23 respondents were excluded due to incomplete answers or incorrect answers to the second screening question. Therefore, the final sample used to examine our proposed framework was 359 cases (80.3% of the total responses).

There were more male respondents (57.1%) than female ones (42.9%). Half of the respondents were aged 18 to 25 years old (50.7%), and more than one-third were between 26 and 35 (34.5%). Regarding the highest education level, most of the respondents hold a bachelor's degree (65.7%). Additionally, 42.3% of the respondents had a monthly income from USD 1001 to 2000. Finally, the majority of respondents frequently used the chatbot services once or twice a month (74.7%). The respondents' demographic information is provided in Table 2.

**Table 2.** Demographic information of the respondents.

| Category | Subcategory | Frequency | Percentage |
|---|---|---|---|
| Having experience with chatbots | Yes | 359 | 100% |
| Gender | Male | 205 | 57.1% |
| | Female | 154 | 42.9% |
| Age | Below 18 years old | 5 | 1.4% |
| | 18 to 25 years old | 182 | 50.7% |
| | 26 to 35 years old | 124 | 34.5% |
| | 36 to 45 years old | 38 | 10.6% |
| | 46 to 55 years old | 8 | 2.2% |
| | Above 55 years old | 2 | 0.6% |
| The highest education level | High school diploma | 47 | 13.1% |
| | Bachelor's degree | 236 | 65.7% |
| | Master's degree | 71 | 19.8% |
| | Doctoral degree | 5 | 1.4% |
| Monthly income | Below USD 500 | 54 | 15.1% |
| | USD 500–1000 | 113 | 31.5% |
| | USD 1001–2000 | 152 | 42.3% |
| | USD 2001–3000 | 28 | 7.8% |
| | Above USD 3000 | 12 | 3.3% |
| Usage frequency | Less than 1 time per month | 45 | 12.5% |
| | 1–2 times per month | 268 | 74.7% |
| | 1–2 times per week | 32 | 8.9% |
| | More than 2 times per week | 14 | 3.9% |

## 5. Data Analysis and Results

Structural equation modeling (SEM) was adapted as the main data analysis method. We followed a two-step approach [99] to examine both the measurement model and structural model.

### 5.1. Measurement Model

Thanks to AMOS software (version 24.0, International Business Machines Incorporation, Armonk, New York, USA), confirmatory factor analysis (CFA) was conducted to

examine the model fit criteria, reliability, and validity of the measurement model. The measurement model was evaluated via five procedures.

First, this study employed the factor loading with a value exceeding 0.6 as the evaluation criterion. If the factor loading for any item is higher than 0.6, this item was retained for further analysis [100]. Table 3 showed that the factor loadings for all latent constructs significantly exceeded the threshold of 0.6. Thus, no item was removed from the scale.

**Table 3.** Reliability and convergent validity of the measurement model.

| Constructs | Items | Factor Loading | *t*-Value | Cronbach's Alpha | Composite Reliability | AVE |
|---|---|---|---|---|---|---|
| System Quality (SYQ) | SYQ1 | 0.846 | - | 0.889 | 0.892 | 0.623 |
| | SYQ2 | 0.848 | 18.254 | | | |
| | SYQ3 | 0.799 | 16.694 | | | |
| | SYQ4 | 0.727 | 14.553 | | | |
| | SYQ5 | 0.717 | 14.273 | | | |
| Information Quality (INQ) | INQ1 | 0.804 | - | 0.913 | 0.913 | 0.601 |
| | INQ2 | 0.795 | 15.706 | | | |
| | INQ3 | 0.813 | 16.198 | | | |
| | INQ4 | 0.745 | 14.427 | | | |
| | INQ5 | 0.724 | 13.904 | | | |
| | INQ6 | 0.793 | 15.656 | | | |
| | INQ7 | 0.750 | 14.535 | | | |
| Service Quality (SEQ) | SEQ1 | 0.855 | - | 0.926 | 0.926 | 0.677 |
| | SEQ2 | 0.859 | 19.680 | | | |
| | SEQ3 | 0.821 | 18.233 | | | |
| | SEQ4 | 0.749 | 15.755 | | | |
| | SEQ5 | 0.815 | 17.983 | | | |
| | SEQ6 | 0.834 | 18.698 | | | |
| Trust (TRU) | TRU1 | 0.822 | - | 0.909 | 0.911 | 0.720 |
| | TRU2 | 0.876 | 18.592 | | | |
| | TRU3 | 0.868 | 18.359 | | | |
| | TRU4 | 0.826 | 17.063 | | | |
| Confirmation of expectations (CON) | CON1 | 0.889 | - | 0.907 | 0.908 | 0.766 |
| | CON2 | 0.880 | 21.174 | | | |
| | CON3 | 0.857 | 20.238 | | | |
| Perceived usefulness (PU) | PU1 | 0.842 | - | 0.887 | 0.888 | 0.664 |
| | PU2 | 0.823 | 16.977 | | | |
| | PU3 | 0.807 | 16.511 | | | |
| | PU4 | 0.786 | 15.896 | | | |
| Satisfaction (SAT) | SAT1 | 0.844 | - | 0.894 | 0.895 | 0.680 |
| | SAT2 | 0.836 | 17.989 | | | |
| | SAT3 | 0.795 | 16.641 | | | |
| | SAT4 | 0.823 | 17.547 | | | |
| Continuance intention (CI) | CI1 | 0.856 | - | 0.880 | 0.881 | 0.712 |
| | CI2 | 0.846 | 18.080 | | | |
| | CI3 | 0.829 | 17.563 | | | |

Second, the Cronbach's alpha of eight constructs ranged from 0.880 to 0.926, and the composite reliability ranged from 0.881 to 0.926 (see Table 3), significantly exceeding

the recommended threshold of 0.70 [100]. Thus, all constructs achieved the ideal internal consistency and reliability.

Third, Fornell and Larcker [101] suggested that the measurement model must meet three following conditions to achieve convergent validity: (1) the factor loadings of all items within the observed variable must be higher than 0.5; (2) the composite reliability for each construct must exceed 0.7; and (3) the average variance extracted (AVE) for each construct must be higher than 0.5. As shown in Table 3, the AVE for each construct exceeded the cut-off value of 0.5 [102], and all latent constructs met the three conditions recommended by Fornell and Larcker [101]. These results confirmed the convergent validity of the measurement model.

Fourth, this study followed the criterion suggested by Fornell and Larcker [101] to assess the discriminant validity of the measurement model, in which the shared correlations between any pair of constructs must be inferior to the square root of the AVE for each construct. Table 4 showed that the highest inter-construct correlation (0.723 between SAT and CI) was lower than the lowest square root of AVE (0.775 for INQ), confirming the acceptable discriminant validity of the instrument.

**Table 4.** The correlation matrix and discriminant validity.

| Construct | SYQ | INQ | SEQ | TRU | CON | PU | SAT | CI |
|---|---|---|---|---|---|---|---|---|
| SYQ | **0.789** | - | - | - | - | - | - | - |
| INQ | *0.585* | **0.775** | - | - | - | - | - | - |
| SEQ | *0.640* | *0.707* | **0.823** | - | - | - | - | - |
| TRU | *0.614* | *0.552* | *0.516* | **0.849** | - | - | - | - |
| CON | *0.526* | *0.479* | *0.532* | *0.662* | **0.875** | - | - | - |
| PU | *0.653* | *0.543* | *0.614* | *0.548* | *0.454* | **0.815** | - | - |
| SAT | *0.720* | *0.657* | *0.669* | *0.718* | *0.694* | *0.663* | **0.825** | - |
| CI | *0.662* | *0.544* | *0.569* | *0.688* | *0.691* | *0.609* | *0.723* | **0.844** |

Note: SYQ = System quality; INQ = Information quality; SEQ = Service quality; TRU = Trust; CON = Confirmation of expectations; PU = Perceived usefulness; SAT = Satisfaction; CI = Continuance intention. The squares root of average variance extracted are represented by the diagonal elements in bold. The correlation coefficients are represented by the italic elements.

Fifth, after the reliability and validity requirements were met, the next step was to evaluate the goodness of fit of the measurement model. This study applied the fit and assessment indicators taken from Bentler and Bonett [103], Bentler [104], Bentler [105], Bagozzi et al. [106], Hu and Bentler [107], and Henry and Stone [108] (see Table 5). The results shown in Table 5 indicate that all indexes exceeded the cut-off values, supporting the acceptable model fit.

Since the current study used self-reported surveys to validate the theoretical model, the responses may be affected by common method bias (CMB). To minimize the effects of the CMB, we made more effort to guarantee the anonymity of respondents. Additionally, a pretest of the measurement items adapted from the previous studies was conducted to improve the internal validity of the research constructs [109]. In addition, two statistical tests were conducted to examine whether the CMB was a severe threat to the current study.

(1) We followed the recommendation of Podsakoff et al. [109] by conducting a Harman one-factor test [110]. All items were included in the exploratory factor analysis (without rotation) using SPSS software version 25.0. The examination results indicated that the first factor accounted for 39.24% of the total variance, which was lower than the threshold of 50% [109,110]. Therefore, the CMB is not a serious problem in our research.

(2) According to Bagozzi et al. [106], the CBM may happen if there is at least one correlation value among the constructs being higher than 0.90. As can be seen in Table 4, the highest correlation value (0.723 for SAT-CI) was considerably below 0.90, confirming that there is no evidence of CMB.

### 5.2. Structural Model

The structural equation modeling (SEM) method was applied to analyze the hypotheses of the structural model. This method allows researchers to examine both the overall fitness and the relationships of the research model [102]. We also used the AMOS 24.0 software for the data analysis.

**Table 5.** The goodness-of-fit indicators for the measurement model and structural model.

| Indicators | Accepted Criteria | Measurement Model | Structural Model | Result | Sources |
|---|---|---|---|---|---|
| $\chi^2/df$ | $\leq 3$ | 1.352 | 1.405 | Good | [106] |
| CFI | $\geq 0.90$ | 0.976 | 0.972 | Good | [106] |
| NFI | $\geq 0.90$ | 0.942 | 0.938 | Good | [106] |
| IFI | $\geq 0.90$ | 0.978 | 0.976 | Good | [104] |
| RFI | $\geq 0.90$ | 0.934 | 0.932 | Good | [103,105] |
| GFI | $\geq 0.90$ | 0.912 | 0.905 | Good | [106] |
| AGFI | $\geq 0.90$ | 0.915 | 0.912 | Good | [103] |
| PGFI | $\geq 0.50$ | 0.765 | 0.761 | Good | [103,105] |
| PCFI | $\geq 0.50$ | 0.783 | 0.781 | Good | [103,105] |
| PNFI | $\geq 0.50$ | 0.778 | 0.775 | Good | [103,105] |
| RMR | $\leq 0.08$ | 0.051 | 0.059 | Good | [107] |
| RMSEA | $\leq 0.08$ | 0.025 | 0.029 | Good | [108] |

Note: $\chi2/df$ = ratio of the chi-square value to the degree of freedom; CFI = Comparative fit index; GFI = Goodness of-fit index; AGFI = Adjusted goodness-of-fit index; IFI = Incremental fit index; NFI = Normed fit index; RFI = Relative fit index; PCFI = Parsimonious comparative fit index; PGFI = Parsimonious goodness-of-fit index; PNFI = Parsimonious normed fit index; RMSEA = Root mean square error of approximation; RMR = Root mean square residual.

Similar to the measurement model, the goodness of fit of the structural model was evaluated via 12 indicators. The results in Table 5 indicate that the structural model achieved the acceptable model fit.

To enhance the internal validity of the structural model, we controlled for the effects of the demographic variables in the structural model analysis, including age, gender, and chatbot usage frequency. The influence of these control variables on the dependent variable (i.e., continuance intention) was not significant (see Figure 2).

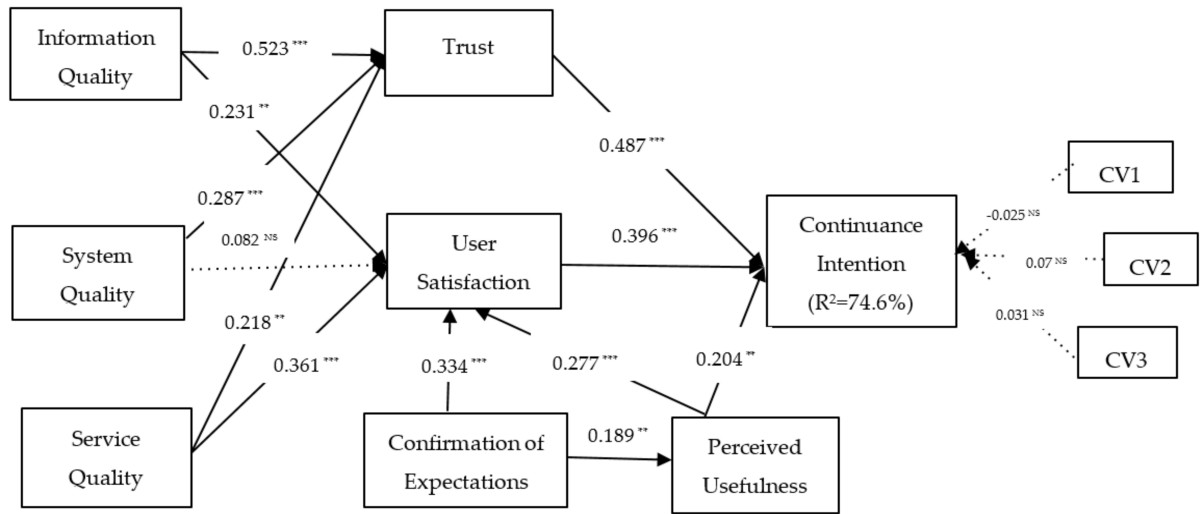

**Figure 2.** The path coefficients of the research model. Note: *** *p*-value < 0.001; ** *p*-value < 0.01; NS: non-significant (*p* = value > 0.05). CVs are control variables; CV1 = Age; CV2 = Gender; CV3 = chatbot usage experience.

As shown in Figure 2 and Table 6, most of the proposed hypotheses in this study were supported, except H2b. In particular, information quality significantly and positively influenced both trust (β = 0.523, *p* < 0.001) and satisfaction (β = 0.231, *p* < 0.01), confirming H1a and H1b. The effect of system quality on trust was significant and positive (β = 0.287, *p* < 0.001), supporting H2a. However, the effect of system quality on satisfaction was insignificant (β = 0.082, *p* > 0.05), rejecting H2b. As predicted, service quality was positively associated with both trust (β = 0.218, *p* < 0.01) and satisfaction (β = 0.361, *p* < 0.001), confirming H3a and H3b. In addition, confirmation of expectations had positive and significant effects on perceived usefulness (β = 0.189, *p* < 0.01), as well as on satisfaction (β = 0.334, *p* < 0.001), supporting H4 and H5. Perceived usefulness also significantly and positively affected satisfaction (β = 0.277, *p* < 0.001); therefore, H7 was confirmed. As expected, satisfaction, perceived usefulness, and trust positively and significantly influenced continuance intention (β = 0.396; 0.204 and 0.487, respectively, *p* < 0.01), confirming H6, H8, and H9. In sum, the research model accounted for 74.6% of the variance in the user's intention to continue using banks' chatbot services.

**Table 6.** The hypotheses testing results.

| Hypotheses | Paths | Standardized Path Coefficients | Support |
| --- | --- | --- | --- |
| H1a | INQ → TRU | 0.523 *** | Yes |
| H1b | INQ → SAT | 0.231 ** | Yes |
| H2a | SYQ → TRU | 0.287 *** | Yes |
| H2b | SYQ → SAT | 0.082 [NS] | No |
| H3a | SEQ → TRU | 0.218 ** | Yes |
| H3b | SEQ → SAT | 0.361 *** | Yes |
| H4 | CON → PU | 0.189 ** | Yes |
| H5 | CON → SAT | 0.334 *** | Yes |
| H6 | SAT → CI | 0.396 *** | Yes |
| H7 | PU → SAT | 0.277 *** | Yes |
| H8 | PU → CI | 0.204 ** | Yes |
| H9 | TRU → CI | 0.487 *** | Yes |

Note: *** *p*-value < 0.001; ** *p*-value < 0.01; NS: non-significant (*p* = value > 0.05).

## 6. Discussion

This study is mainly focused on integrating DeLone and McLean's ISS model [59], the expectation confirmation model [43], and trust to shed light on the issue of continuance intention regarding banks' chatbot services. Several key findings from the analysis results are discussed as follows.

First, the relationship between information quality and trust is supported, which is similar to the findings of Lee and Chung [85], Gao and Waechter [81], and Ofori et al. [111]. The satisfaction of users is also positively affected by information quality. This result is in line with the D&M ISS models [59,60] and several previous studies (e.g., [42,65,73]). In addition, among the three dimensions of the D&M ISS model, information quality has the strongest effect on trust (β = 0.523). Our findings, thereby, emphasize the important role of information quality in enhancing users' satisfaction and especially trust towards banks' chatbot services. Acquiring needed information and support are the two major motivations for users to use chatbots [112]. These are justifiable in the context of banking when interests, exchange rates, and other important indexes constantly change, and complex banking procedures often struggle with users. Hence, if the chatbots provide users with relevant, precise, and updated information, their financial decisions will be made quickly and correctly. Once users perceive chatbots as trustworthy, they will feel more satisfied [75].

Second, the influences of service quality on both trust and satisfaction are also significantly positive. Importantly, service quality is the strongest predictor of user satisfaction ($\beta$ = 0.361). These findings prove the validity of the long-established perspectives in marketing studies that service quality remains one of the key determinants of satisfaction [113]. Similar findings can be found in the existing IS studies (e.g., [42,57,81,84,111]). It can be inferred from these results that if the bank's chatbots provide prompt responses, relevant suggestions, and individualized attention to users, their satisfaction and trust could be enhanced. In fact, instead of queuing and waiting for advice from staff when using human-staffed services, banks' customers select chatbot services as a time-saving alternative. Therefore, if chatbots cannot guarantee promptness and personalization, users may suspect that banks cannot provide high-quality services, which can decrease their trust and satisfaction.

Third, the relationship between system quality and user satisfaction is not significant, which is incoherent with the D&M ISS model [59] and findings of some existing studies in mobile payment and e-government systems (e.g., [57,75,76,84]). One possible explanation can be that using chatbot services does not require much effort from users. They can start conversations with chatbots by simply typing messages or using their voices, leading to system quality becoming less important than service quality and information quality in the relationship with satisfaction. This result also reinforces the study of Ashfaq et al. [42], who only considered information quality and service quality within the D&M ISS model [59] as the two predictors of satisfaction. Unlike satisfaction, the effect of system quality on trust is supported, which is in line with the findings of Zhou [76] and Gao et al. [75]. This reflects the fact that chatbot users are worried about information disclosure and data-stealing. Compared to some developed economies, the legal frameworks regarding consumers' privacy protection in online environments in many developing countries and emerging markets, such as Vietnam, have not been strong enough. Banks in Vietnam hardly ensure comprehensive solutions to information disclosure. Hence, providing good system quality in terms of reliability and security has a significant role in enhancing users' trust in chatbot services.

Fourth, confirmation of expectations is a significant driver of users' satisfaction, perceived usefulness, and continuance intentions. These findings strongly support the postulate of the post-adoption model of IS continuance (i.e., ECM). Bhattacherjee [43] posited that the initial expectation of IS users might change based on the post-adoption experience and the confirmation of the updated expectation should be validated as the cognitive beliefs influencing the consequent processes (i.e., perceived usefulness, satisfaction) to address the users' continuance intentions. Our results are in line with many previous empirical studies in different contexts (e.g., [45,57,89,98]). This means that if users find the actual performance of chatbots good enough to meet their expectations, they will perceive chatbots as more valuable and be satisfied with them, which could result in continuance intentions.

Fifth, our research also pinpoints that perceived usefulness is an essential antecedent of user satisfaction and continuance intention, which validates the original findings of Bhattacherjee [43]. This implies that if users perceive banks' chatbots as beneficial to them, they will be more satisfied and more likely to continue using them in the future. Furthermore, the significant effect of satisfaction on continuance intentions reinforces the extant marketing literature that user satisfaction is the critical determinant of continuance intentions. This means that the more satisfied banks' users are with the chatbot, the more likely they are to continue to use it.

Finally, trust is found to have the strongest effect ($\beta$ = 0.487) on continuance intention. This finding highlights the crucial role of trust in predicting users' intentions to continue using bank's chatbots, which is a new finding in chatbot-related studies. Chatbots are programmed to communicate with users through online chat conversations, thereby involving potential uncertainties and risks. Trust could reduce users' perceptions of these risks, worries, and uncertainties [92,94]. Thus, users who believe banks' chatbot services to be highly trustworthy will be more willing to continue using them in the future. The

strongest impact of trust on continuance intention is also reasonable for the finance-related contexts, such as banking, in which customers tend to continue using specific services only if they trust them. Our result also helps further the existing findings in other contexts, such as Fintech [8] or mobile payment [96,97].

## 7. Implications

### 7.1. Theoretical Implications

This study contributes to the progression of the theoretical foundation related to chatbot services and IS continuance in several ways. First, this study is one of very few attempts to explore the key determinants affecting users' continuance intentions regarding chatbots from the perspectives of the ECM [43], the D&M ISS model [59], and the trust concept. Several researchers have advocated using more pertinent theories to examine the information technology users' continuance intentions by using traditional models such as the TAM, UTAUT, and TPB [114,115]. Although the most recent study regarding chatbot e-services [42] also relied on the ECM and the D&M ISS model, it failed to consider the determining effects of trust, system quality, confirmation of expectations, which were demonstrated as essential elements of our study. By proposing and empirically testing the integrated model, which incorporates all variables from these two models and the trust concept, our findings are expected to offer both academics and practitioners a deeper insight into the antecedents of continuance intentions towards information systems. In addition, most relationships among these constructs in the research model were supported, which is justifiable for why we used the D&M ISS model jointly with the ECM as the theoretical basis. In fact, this combined model has a high explanatory power, explaining 74.6% of the variance in continuance intention towards chatbots. These results provide the impetus for academics to simultaneously consider the ECM and the D&M ISS model in future studies on users' continuance intentions regarding other information systems.

Second, although trust has proven to be one of the essential drivers of users' continuance intentions in many contexts (e.g., [8,97,116]), no research has thus far investigated its role in the context of chatbots. By empirically demonstrating that satisfaction, perceived usefulness, and trust are significant determinants of chatbot users' continuance intentions, our study sheds new light on the role of trust in strengthening the users' willingness to continue using chatbots. In existing studies (e.g., [42,43,77,91]) and the current study, the proven roles of user satisfaction and perceived usefulness in determining user's continuance intentions suggest that these two constructs should not be excluded from future studies on chatbot services and on information technology continuance.

Third, this study enhances our understanding of consumers' behaviors in the context of banking services. The banking industry is undergoing a transformation towards smart, innovative banking and is currently focusing on improving customers' experiences by leveraging the support of new technologies and Fintech. Transforming the banking experience indeed leads to changes in customers' behaviors [20]. Although the banking sector benefits from implementing the chatbot services, the empirical investigation of users' behavioral intentions towards banks' chatbot services in the post-adoption stages remains sparse and limited. By borrowing the theoretical lens of the ECM and ISS D&M model, this study will provide academics and practitioners with an in-depth understanding of customers' reactions in the post-adoption stage towards not only banks' chatbots but also other relevant banking services.

### 7.2. Practical Implications

As the implementation of chatbot services is beneficial for banks and their customers, the findings of this research can serve as a reference and valuable guideline for financial institutions in formulating practical solutions to promote customers' usage continuance. Our study indicates that banks' customers tend to continue using chatbot services only if they are trustworthy and useful, and customers' needs are satisfied. Therefore, banking ser-

vice providers must pay close attention to the three quality aspects of chatbot services (i.e., information quality, service quality, system quality) and users' confirmation of expectations.

First, since information quality acts as one of the vital signals for both users' trust and satisfaction and exerts the highest effect on users' trust, banking service providers must provide chatbot users with precise, reliable, personalized, relevant, and up-to-date information. More importantly, the information provided to customers via chatbot services must be highly related to their current needs or concerns. In light of this, chatbots must be programmed to optimize and offer appropriate suggestions to users. Essential banking information, such as interest rates, exchange rates, credit cards, and credit-granting processes, must be frequently updated to provide chatbot users with the most precise support. If banks' customers cannot receive the needed information from chatbot systems or the quality of information is low, customers' continuance intentions will be reduced. Furthermore, low-quality information may waste users' effort and time spent on such useless works [84] and increase information-processing costs, which, in turn, reduces their satisfaction and trust in both chatbots and service providers.

Second, service quality is also another critical driver of users' satisfaction and trust, indicating that banks need to offer accurate information and prompt responses to users' queries at the same time via chatbot systems. To speed up the chatbot's responses to customers, banking service providers should set up sets of often-asked familiar keywords and prepare various scenarios to respond promptly. Additionally, all message histories between customers and chatbots can be saved and referred to later; service providers should program the chatbot systems to scan through chat histories to respond promptly to customers. Chatbots must be programmed to offer alternative solutions connecting with direct employees for timely support in case of unavailable answers. A shorter waiting time is necessarily considered to satisfy the user's experience and boost re-usage intentions significantly. In addition, service providers are suggested to provide personalized chatbot-based services to users. For example, if a bank's chatbot interacts with customers by their names, customers may feel as natural as talking to an actual employee. By doing so, chatbots can give users a sense of familiarity, trust and alleviate uncertainty and worries [94]. Once banks provide high-quality chatbot services to customers, they can reap many benefits, such as a good reputation and positive image [42].

Third, we found that system quality is the strongest predictor of trust among the three elements of the IS success model. This finding highlights the users' concerns and requirements for system quality, which strongly affects their trust in chatbot services. Therefore, it is suggested that banks offer chatbot systems with a well-designed, stable, and attractive interface to attract users and make them believe in suppliers' ability to offer a good service. Additionally, service providers should also develop chatbot systems catering to various electronic devices, such as Android, the IOS operating system for mobile, and Windows for computers, to ensure that users can access and interact with chatbots wherever they need [81]. Importantly, banks' chatbots must be programmed to offer 24/7 support to their customers whenever users need them. In addition, in emergent markets, such as Vietnam, banks' customers are more likely to be worried about the security of the chatbot systems because the legal protection towards consumer's privacy has not been strong enough. Therefore, chatbot systems' ability to secure users' data and prevent data loss or personal information disclosure is crucial in building and enhancing users' trust. For this reason, banking service managers should carefully take the systematic risk and users' privacy into consideration when developing chatbot systems.

Fourth, for the positive effect of confirmation of expectations on satisfaction, banks should recognize their customers' expectations from the chatbot services and fulfill them. Since customers' expectations about services change over time [43], banking marketers are well-advised to understand and update them more frequently. Notably, by interviewing customers, sending out survey forms, and encouraging customers to give their feedback about the performance of chatbot services and their experience with chatbots, banking marketers can obtain objective views of chatbot quality and their customer' expectations.

Understanding customers' expectations is the first and essential step for banks to provide timely solutions to satisfy them. Hence, to promote customers' continuance intentions, their expectations must be met or surpassed.

Fifth, given the critical role of perceived usefulness in the relationship with satisfaction and continuance intention, service providers should ensure that banks' chatbot services are error-free because the service failures may prevent customers from obtaining what they are seeking, leading to users' dissatisfaction. Banks should also predict the common questions or inquiries from users and then program chatbots to finish their tasks efficiently. Additionally, the interactions between users and chatbots should be efficient and straightforward.

## 8. Conclusion and Research Limitations

While chatbot services in the financial sector have received much scholarly attention recently, a search of available databases has shown that no such research has been conducted in Vietnam so far. In the current digital transformation era, AI-enabled technologies, such as chatbots, provide a tool to maximize customer value, enhance customer loyalty, and sustain competitive advantages. With the wide usage of chatbots in delivering services, banks may decrease personnel costs, transactions costs, enhance their customer experience, and increase efficiency. The essential role of chatbots in providing personal and online services is undeniable in some external events, such as the current COVID-19 pandemic outbreak and beyond, which restricts face-to-face communications. By combining the D&M ISS model, the ECM, and the trust concept, this study investigated the determinants of users' continuance intentions towards banks' chatbot services in Vietnam. The findings of our study suggested that banking managers need to leverage factors influencing users' satisfaction, trust, perceived usefulness, and continuance intentions towards banks' chatbot services to develop action plans which contribute to sustainable developments and competitive advantages of banks.

Although the research procedure of this study was as rigorous as possible, our study still has the following limitations. First, the results of this study were analyzed based on the small sample size of the banks' chatbot users through the convenience sampling technique; the respondents and results, therefore, are not generalized and representative of the entire banks' chatbot users in Vietnam. Additionally, this study collected data by using the self-reported survey method. Even though common method bias was not a serious problem, as demonstrated before, it is always a concern [109]. Thus, future studies should consider other types of approaches, such as the experimental method, to enhance the quality of respondents.

Second, our results only reflect the chatbot usage in a single context (i.e., Vietnam), an emerging market. The differences across countries, areas, cultures, or country-development levels may also influence our findings. Therefore, to strengthen the systematization of the current study, future studies can compare the current results with those from other countries with different cultural backgrounds (e.g., Western countries) and levels of development (e.g., developed markets vs. this emergent market). In addition, replicating this study in different contexts or industries and comparing the results with each other are also encouraged.

Third, the research model in the study was formulated by integrating the D&M ISS model and the trust concept into the ECM to identify the key determinants affecting chatbot users' continuance intentions. Although the constructs included in this study are relevant to continuance intentions and fit the initial research purposes, it is worthwhile to include other essential constructs that could predict continuance intentions. We suggest that future studies could extend the current research model by including personality-related concepts, such as self-efficacy, technology readiness, and compatibility, which may further explain continuance intentions. In addition, this study omitted the effect of motivational factors on user behavioral intentions. The follow-up study should apply motivation-related theories,

such as self-determination theory, a well-known theory in psychology, to explore the impact of internal motivation on chatbot users' continuance intentions.

Finally, although our primary purpose was to focus on the intention to continue using the banks' chatbots, the research on whether intention can serve as a proxy for behavior is ongoing [117]. Considering that research on the IS continuance aims to boost an actual re-usage behavior, it should be more effective to measure actual re-usage behavior instead of intention [115]. However, the link between intention to continue using chatbots and actual continuance usage has not been investigated. This remains a significant gap and opens the opportunity for future research to bridge.

**Author Contributions:** Conceptualization, D.M.N. and Y.-T.H.C.; methodology, D.M.N., Y.-T.H.C. and H.D.L.; validation, D.M.N. and Y.-T.H.C.; formal analysis, D.M.N. and H.D.L.; investigation, D.M.N. and H.D.L.; writing—original draft preparation, D.M.N. and H.D.L.; writing—reviewing and editing, D.M.N. and Y.-T.H.C.; visualization, D.M.N. and Y.-T.H.C.; supervision, Y.-T.H.C. All authors have read and agreed to the published version of the manuscript.

**Funding:** This research received no external funding.

**Institutional Review Board Statement:** Ethical review and approval was not required for this study on human participants in accordance with the local legislation and institutional requirements.

**Informed Consent Statement:** Written informed consent from the patients/participants was not required to participate in this study in accordance with the national legislation and the institutional requirements.

**Data Availability Statement:** The data presented in this study are available on request from the corresponding author. The data are not publicly available due to assured participant confidentiality.

**Conflicts of Interest:** The authors declare no conflict of interest.

## Appendix A

**Table A1.** Items used to measure research constructs.

| | Measurement Items | Source |
|---|---|---|
| | **System Quality** | |
| SYQ1 | This bank's chatbot system is easy to use | |
| SYQ2 | This bank's chatbot system is user-friendly | |
| SYQ3 | Using this bank's chatbot system does not require much effort | [77] |
| SYQ4 | I could use bank's chatbot system whenever and wherever I want | |
| SYQ5 | Using this bank's chatbot system is comfortable | |
| | **Information Quality** | |
| INQ1 | Information provided by this bank's chatbot is reliable | |
| INQ2 | Information provided by this bank's chatbot is accurate | |
| INQ3 | Information provided by this bank's chatbot is easy to understand | |
| INQ4 | Information provided by this bank's chatbot is up-to-date | [77] |
| INQ5 | Information provided by this bank's chatbot is in an eye-catching format | |
| INQ6 | I have received the sufficient information from this bank's chatbot | |
| INQ7 | This bank's chatbot provides me with necessary information on time when I need it | |

**Table A1.** *Cont.*

| | Measurement Items | Source |
|---|---|---|
| | **Service Quality** | |
| SEQ1 | This bank's chatbot provides me with the exact and appropriate solution to my requirements | [98] |
| SEQ2 | This bank's chatbot provides me with an instant response | |
| SEQ3 | This bank's chatbot gives me the personalized attention | |
| SEQ4 | The interface of this bank's chatbot is modern-looking | |
| SEQ5 | This bank's chatbot has a great interface to communicate my needs | |
| SEQ6 | This bank's chatbot has visually attractive materials | |
| | **Trust** | |
| TRU1 | I believe that this bank's chatbot is trustworthy | [82] |
| TRU2 | I do not doubt the honesty of information provided by this bank's chatbot | |
| TRU3 | I feel assured that this bank's chatbot service has ability to protect users | |
| TRU4 | Overall, I trust in this bank's chatbot | |
| | **Confirmation of Expectations** | |
| CON1 | My experience with this bank's chatbot was greater than my expectations | [43] |
| CON2 | The service level provided by this bank's chatbot was greater than what I expected | |
| CON3 | In general, most of my expectations from using this bank's chatbot were confirmed | |
| | **Perceived Usefulness** | |
| PU1 | Using this bank's chatbot helps me to complete tasks more promptly | [91] |
| PU2 | Using this bank's chatbot increases my productivity | |
| PU3 | Using this bank's chatbot helps me to perform many things more conveniently | |
| PU4 | For me, this bank's chatbot is useful in terms of supporting my requests | |
| | **Satisfaction** | |
| SAT1 | This bank's chatbot has met my expectations | [77] |
| SAT2 | This bank's chatbot efficiently fulfilled my needs (e.g., seeking information, making transaction) | |
| SAT3 | I am pleased with support from this bank's chatbot | |
| SAT4 | Overall, I am satisfied with this bank's chatbot | |
| | **Continuance Intention** | |
| CI1 | I intend to continue using this bank's chatbot in the future | [43] |
| CI2 | I will always try to use this bank's chatbot when I have the need | |
| CI3 | I would strongly recommend this bank's chatbot to other persons | |

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
