# Peer review of "Determinants of Continuance Intention towards Banks’ Chatbot Services in Vietnam: A Necessity for Sustainable Development"

_sustainability, doi:10.3390/su13147625_

Round 1
Reviewer 1 Report
The paper provides a novel idea about determinants of continuance intention towards bank's chatbot service in Vietnam: A necessity for sustainable development. In the future, the insights of such a study could be helpful for the banking industry. Moreover, the references used in this study give the idea that the development of the conceptual framework was done with thorough reading.
The theories used are appropriate for consumer intentions. ECM and ECT are well-accepted theories concerning continuance intention studies. Hence, literature related to the two theories and context-specific variables is appropriate to understand this phenomenon.
The methodology section can be further elaborated on other than the table-1. More information can be useful for the readers in regards to where that study was conducted and will the results be generalized for other countries or industries. Moreover, the results are adequate. The tables and figures justify the overall analysis of this study.
The discussion section needs to be reduced a little as it is long for my liking. Further, the theoretical and practical contributions section needs to be elaborated. The idea is right, but it needs more explanation for academics and practitioners.
The idea of the paper is interesting. Authors can go through further to remove any errors or mistakes related to grammar. Overall, the reader can understand the concept, but further improvements are always helpful in making the paper more reflective.
Author Response
Dear Reviewer,
Thank you very much for your encouraging and inspiring feedback on our work and for your constructive and helpful comments that have definitely improved the paper a great deal. We had studied all of your comments carefully and tried to incorporate all of them into the current version of this article. Please find more detailed descriptions of how we did that in attached pdf file, namely “Response to Reviewer’1 comments”. We hope that you will be satisfied with our effort.

Reviewer 2 Report
Dear Author(s),
Please find below my recommendations about your manuscript proposal.
First of all, I appreciate your work and the general organisation of your article.
You have a logic flow and the paper can be read in a logic manner.
Most of the sections are well-written and the arguments are accessible to the readers.
As I read the article, I really appreciated the general organization, the clear way of presenting the methodology and the research results. Also, each stage of the research is very well argued and substantiated.
My main remarks are about the Introduction section and the Practical Implications sub-chapter.
In the Introduction section you discuss about the main aspects regarding the chatbots and their implications on clients and companies.
I recommend you to better describe the general context of your research by including some important aspects about fintechs (here you could cite https://doi.org/10.15240/tul/001/2021-2-007 because this article is about fintechs and customers' perceptions on them), https://doi.org/10.3390/sym11121449 (this article is about the expected benefits for the banks' clients), https://doi.org/10.1108/K-05-2020-0259 (this article is about the implications of social cryptocurrencies on sustainable development).
In section "7.2. Practical Implications" you say: "To summarize, according to our little knowledge, this is the first study on chatbot 768 service in Vietnam’s banking sector."
Please revise this sentence and include a small paragraph where you argue this point by specifying that you looked for this type of articles in the specialized research databases, but you didn't find specific research on Vietnam.
This way, you have an objective reason to present your own practical implications based on your findings.
Dear Author(s),
Please consider all the above remarks as being constructive recommendations in order to improve the general quality of your manuscript proposal.
Kind Regards!
Author Response
Dear Reviewer,
Thank you very much for your encouraging and inspiring feedback on our work and for your constructive and helpful comments that have definitely improved the paper a great deal. We had studied all of your comments carefully and tried to incorporate all of them into the current version of this article. Please find more detailed descriptions of how we did that in attached pdf file, namely “Response to Reviewer 2's comment”. We hope that you will be satisfied with our effort.
